# Refine Knowledge of Large Language Models via Adaptive Contrastive Learning

**Yinghui Li**[1,*] **Haojing Huang**[1,*], **Jiayi Kuang**[2,*], **Yangning Li**[1], **Shu-Yu Guo**[1]
**Chao Qu**[3,], **Xiaoyu Tan**[3,†] **Hai-Tao Zheng**[1,4,†], **Ying Shen**[2,†], **Philip S. Yu**[5]
[1]Tsinghua University, [2]Sun-Yat Sen University
[3]INFLY TECH (Shanghai) Co., Ltd., [4]Peng Cheng Laboratory
[5]University of Illinois Chicago
{liyinghu20, hhj23}@mails.tsinghua.edu.cn

## Abstract

How to alleviate the hallucinations of Large Language Models (LLMs) has always been the fundamental goal pursued by the LLMs research community. Looking through numerous hallucination-related studies, a mainstream category of methods is to reduce hallucinations by optimizing the knowledge representation of LLMs to change their output. Considering that the core focus of these works is the knowledge acquired by models, and knowledge has long been a central theme in human societal progress, we believe that the process of models refining knowledge can greatly benefit from the way humans learn. In our work, by imitating the human learning process, we design an Adaptive Contrastive Learning strategy. Our method flexibly constructs different positive and negative samples for contrastive learning based on LLMs' actual mastery of knowledge. This strategy helps LLMs consolidate the correct knowledge they already possess, deepen their understanding of the correct knowledge they have encountered but not fully grasped, forget the incorrect knowledge they previously learned, and honestly acknowledge the knowledge they lack. Extensive experiments and detailed analyses on widely used datasets demonstrate the effectiveness of our method.

## 1 Introduction

Based on the massive training corpora and a large number of training resources from the real knowledge of the human world, Large Language Models (LLMs) store many general and domain-specific knowledge and show excellent performance in many natural language processing tasks (Wang et al., 2024; Zhang et al., 2024a). LLMs are equipped with excellent command comprehension, logical reasoning, and text generation capabilities. As chat assistants, LLMs help users realize many tasks in their work, study, and daily life, greatly enriching the user experience and satisfying their needs (Bansal et al., 2024; Asadi et al., 2024; Guo et al., 2024). However, LLMs-based chat assistants, although greatly helping users, show hallucination problems in more and more tasks (Chen et al., 2024; Zhang et al., 2024e;d). They encounter factual errors in their answers, sometimes even fabricate non-existent knowledge, or contradict the user's original intention when entering instructions (Mu & Lim, 2024; Lv et al., 2024). Some of these hallucinatory errors, such as fabricating false documentaries, are difficult for the user to recognize directly, but they can have serious consequences when used and are likely to **reduce the user's trust in the AI assistant.**

When the LLM hallucinates in its reply, provides factual errors, or fabricates false knowledge, we can assume that the LLM lacks the relevant knowledge for the problem and does not have the ability to respond accurately (Li et al., 2024; Cheng et al., 2024; Zhang et al., 2024c). From the perspective of human knowledge needs, for questions beyond the scope of LLM's knowledge, we receive an honest response like "I don't know" rather than a fabricated, plausible-sounding answer. This will be more conducive to utilizing AI to get the valid information that users need, as users can actually

---

*indicates equal contribution.
†Corresponding authors: Hai-Tao Zheng, Ying Shen, and Xiaoyu Tan.

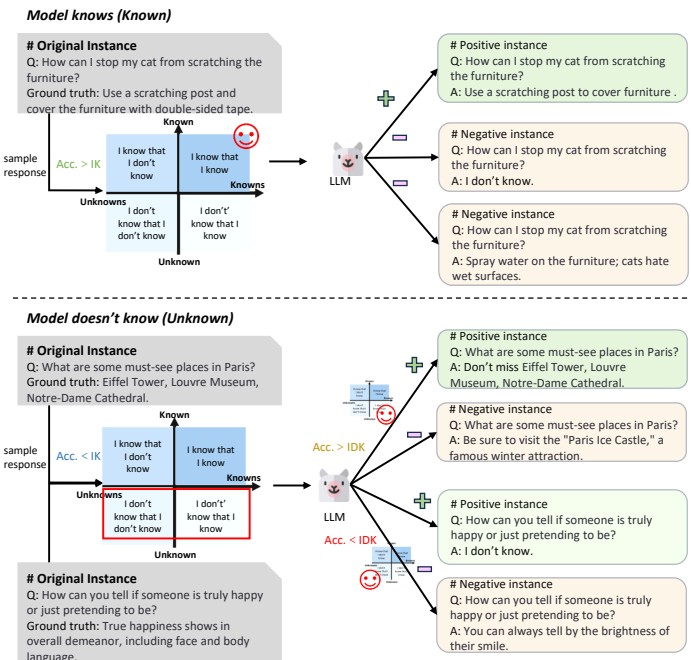

Figure 1: The illustration of our adaptive contrastive learning. Based on the multiple sampling responses of the model, we represent the knowledge regions with different mastery capabilities, and construct different positive and negative examples of adaptive contrastive learning to help the model better consolidate what is known.

achieve what they want by providing more knowledge about the initial question, or by asking for additional help.

Therefore, to get an honest LLM assistant with higher creditworthiness, the LLM must have two abilities: the LLM not only has to be **able to choose to refuse to answer** when faced with an unknown question but also has to acquire the ability to **correctly distinguish whether it really knows the knowledge related to a question or not,** which means the model needs to be aware of what it knows and what it does not (Zhang et al., 2024c). The LLM's ability to perceive knowledge can be represented by the knowledge quadrants (Yin et al., 2023b). The knowledge quadrant is divided into four quadrant regions as shown in Figure 1, Known Knowns, Known Unknowns, Unknown Knowns, and Unknown Unknowns, where the vertical axis represents the model's perception of itself (i.e., the model's thought of what it knows or does not know). The horizontal axis represents the model's actual knowledge mastery (i.e., whether or not the model actually masters a particular piece of knowledge). Thus, knowledge in the first quadrant "Know Knowns", represents that the AI knows that it knows, while knowledge in the forth quadrant "Unknown knowns", represents that the AI does not know that it knows, and the same can be inferred for the other quadrants.

Ideally, the LLM gives the correct response to the question, which falls into "Known Knowns", whereas when the LLM is faced with the hallucination problem, it does not realize that it does not know the relevant knowledge and gives the wrong response, which falls into the "Unknown Unknowns" quadrant. The horizontal axis represents the LLM's specific knowledge mastery, which is limited by the LLM's own training process and knowledge capability. Aiming to mitigate the hallucination problem and meet the needs of the honest LLM system, we pay more attention to the vertical "Known" ability. We believe that providing a correct response to known questions and refusing to respond to unknown questions, as outlined in the two quadrants above, are both acceptable behaviors for an honest LLM. Thus, our goal in mitigating the LLM hallucination problem is to have more input instructions and responses that can be moved **from the lower two quadrants, into the upper two quadrants**.

Research on methods to mitigate the LLM hallucination problem and improve its honesty based on knowledge representation faces the following problems:

1. **The design of knowledge boundaries is relatively simple and much knowledge is not utilized.** In order to help LLMs better represent the knowledge known to the model and the knowledge unknown to the model, early studies such as Zhang et al. (2024c) and Cheng et al. (2024) would design knowledge delineation methods to divide relevant questions into known and unknown, and then further teach the model to answer honestly through fine-tuning. However, these methods of knowledge boundary delineation are simplistic and ignore much of the critical knowledge that the model is uncertain about and does not utilize well. As a result, while the model may refuse to answer questions it does not know, **this reduces the model's confidence and causes it to respond only when it is very certain**, which greatly affects the **validity** of its responses.

2. **Lack of learning from the model's incorrect replies.** In the early work, most of the focus was on the model's known knowledge and its correct responses, as well as the fine-tuning effect on the model, but it did not consider the use of the model's incorrect responses (Cheng et al., 2024). Inspired by the process of human knowledge learning and thinking, we believe that the model not only needs to learn correct knowledge, should also recognize its own mistakes from incorrect replies. By learning from this incorrect feedback, the model can enhance its understanding and representation of knowledge by avoiding incorrect reasoning and forgetting related incorrect knowledge in subsequent learning.

Therefore, we propose **Adaptive Contrastive Learning**, which refine the boundaries of an LLM's knowledge representation to align with real-world human knowledge needs. This approach enables the model to **maintain known knowledge, consolidate uncertain knowledge, and forget unknown knowledge.** We introduce an innovative knowledge boundary design scheme featuring an upper threshold, I Know Rate ($IK$), and a lower threshold, I Don't Know Rate ($IDK$), to delineate knowledge representation. Using well-annotated Question Answering pairs, we input these into the LLM, sample responses multiple times, and calculate the response accuracy. Responses with accuracies greater than or equal to $IK$ indicate the model knows what it knows, those between $IK$ and $IDK$ suggest uncertainty, and those less than $IDK$ imply unknown knowledge. Based on these knowledge quadrants, we construct comparative learning data, designing adaptive positive and negative samples for each representation. This includes learning correct boundary knowledge, addressing incorrect responses, and reinforcing rejected answers. To optimize the LLM using these samples, we implement an adaptive contrastive learning strategy with tailored loss functions for different knowledge representations, while fine-tuning the model's generation capabilities. By maximizing the distance between negative samples and minimizing the distance between positive samples, the model learns accurate knowledge while discarding inaccuracies, thereby enhancing its confidence and reliability. Our contributions are as follows:

- We propose a new knowledge representation method for LLMs, which assists the model in better refining its own knowledge scope, enhances the model's honesty and alleviates the model's hallucination problem through a new knowledge boundary division.

- We design an Adaptive Contrastive Learning strategy, through which the model can maintain its known knowledge, consolidate the known but uncertain knowledge, and forget the unknown knowledge, which improves the validity and honesty of the model's responses.

- We conduct experiments and analyses on various advanced LLMs and test them on both in-distribution and out-of-distribution data. The experimental results show that our approach achieves the highest Truthful rate, verifying the effectiveness of our proposed Adaptive Contrastive Learning strategy.

## 2 RELATED WORK

### 2.1 FINE-TUNING LLMS WITH HUMAN KNOWLEDGE

Large language models learn extensive knowledge from human society and carry out various tasks based on this knowledge (Burns et al., 2023; Ouyang et al., 2022). Among mainstream techniques, supervised fine-tuning (SFT) aligns LLM with real-world human knowledge, playing a critical role in fulfilling the tasks expected of LLM by humans. After gathering a large annotated corpus for natural language processing, researchers fine-tune pre-trained language models, enabling LLMs to

learn from the carefully selected and annotated human knowledge and apply it to unseen tasks (Wei et al., 2022; Sanh et al., 2022). During the SFT process, enhancing model performance can be achieved by expanding the size of fine-tuning data (Chung et al., 2022) and gathering complex data across various domains and tasks (Shen et al., 2023; Longpre et al., 2023), however, it may sacrifice generality by focusing on specific domain fine-tuning. Beyond different data, instruction tuning methods design various instruction-response pairs and enable models to learn response patterns and content more in line with human instruction expectations, making LLMs more truthful, credible, and helpful (Liu et al., 2024b; Muennighoff et al., 2024). However, LLMs in instruction tuning are not robust to changes in instruction texts, often generating inconsistent answers to slightly rephrased instructions (Liu et al., 2024a). Yan et al. (2024) propose a contrastive instruction tuning approach that maximizes similarity between semantically equivalent instruction pairs and minimizes similarity between semantically distinct pairs, significantly enhancing robustness in instruction fine-tuning.

## 2.2 MITIGATING HALLUCINATION WITH LLM'S KNOWLEDGE

LLMs accumulate substantial knowledge during training and fine-tuning, yet unavoidably encounter data compression challenges during training, where the training data size is several times larger than the model's parameters (Zhu et al., 2023). This makes it challenging for LLMs to fully restore the original knowledge, leading to hallucinations (Gekhman et al., 2024). Moreover, when the model lacks sufficient information to respond to questions, it tends to guess, producing inaccurate outputs (Mündler et al., 2024; Zhang et al., 2024d). Hallucinations typically involve logical fallacies (errors in reasoning by the model) (Mu & Lim, 2024), factual errors (when the model confidently asserts non-existent facts) (Lv et al., 2024; Li et al., 2024), and data-driven biases (when certain data prevails, potentially skewing the model's output in certain directions) (Sarkar et al., 2024; Zhang et al., 2024b). Numerous studies focus on enhancing the models' utilization of knowledge representation to mitigate hallucination issues. Asai et al. (2024); Niu et al. (2024) employ the RAG approach, bolstering knowledge retrieval and self-feedback, thereby alleviating the model's hallucinations. Chuang et al. (2024) propose a decoding strategy, which contrasts the differences in tokens' logits obtained from different layers for reducing hallucinations. Cheng et al. (2024); Zhang et al. (2024c) are also exploring the issue of model honesty, asserting that when the model can honestly respond "I don't know" to unknown knowledge, it enhances the model's credibility.

## 3 METHODOLOGY

### 3.1 KNOWLEDGE BOUNDARIES FOR MODELS

Different question-answering datasets encompass various types of knowledge and expressions, making it challenging to assess whether models truly possess specific knowledge due to varying confidence levels. To enable models to better identify their knowledge boundaries and determine their ability to answer related questions, we establish two thresholds: the upper threshold $IK$ (I Know rate) and the lower threshold $IDK$ (I Don't Know rate). These correspond to three knowledge types: *the model knows it knows (knowns-known knowledge), the model does not realize it knows (unknowns-known/uncertain knowledge), and the model does not know it does not know (unknowns-unknown knowledge).*

Following previous studies Cheng et al. (2024); Zhang et al. (2024c), we filter questions from public datasets and query LLMs multiple times to sample responses, calculating accuracy as a measure of the model's confidence in each question. The number of sampled responses and the thresholds $IK$ and $IDK$ serve as hyperparameters. As illustrated in Figure 2, when confidence exceeds $IK$, the model is deemed to fully master that knowledge. If confidence is between $IK$ and $IDK$, the model has the knowledge but may not recognize it, leading to occasional incorrect answers. Confidence below $IDK$ indicates a lack of knowledge, resulting in low correct response rates. By defining these knowledge boundaries, we will construct contrastive learning data and adaptively tune the model based on these knowledge types. This process aims to help the model retain known knowledge, reinforce uncertain knowledge, and forget unknown knowledge. We aspire for adaptive contrastive learning instruction tuning to **shift more knowledge from the third and fourth quadrants to the first and second quadrants**, enhancing model honesty while ensuring helpful responses.

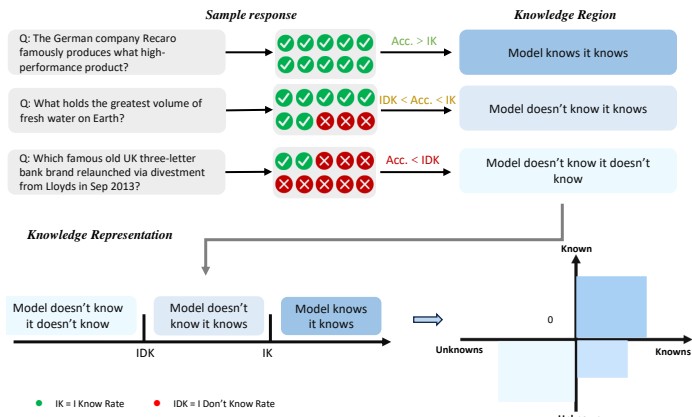

Figure 2: The illustration of the knowledge boundaries and the sample response.

We choose the widely used mainstream knowledge-sensitive open-domain question-answering dataset, TriviaQA, based on which we collect model responses. To determine whether the model's responses are correct or not, we select lexical matching as an automatic evaluation metric to assess whether the model's output is correct or not. By calculating the consistency rate between the model's responses and the human-labeled responses in the dataset, we consider that the model correctly answers the question when it reaches approximately 90%.

## 3.2 CONSTRUCTION OF CONTRASTIVE LEARNING DATA

Based on the knowledge boundary delineation in Section 3.1, we have collected knowledge data of what the model knows and what it does not know, which corresponds to the right and the left quadrants of the knowledge, respectively. We further perform contrastive learning data construction for the knowledge, as shown in Figure 3.

### 3.2.1 QUESTIONS THAT THE MODEL KNOWS

In Section 3.1, the accuracy of the question greater than the upper threshold $IK$ falls in the first quadrant of the knowledge of "what the model knows that it knows". For a given question, the correct response of the corresponding model is designed as a positive instance. There are two types of negative instances: one is the model's response "I don't know", which can be pushed farther away in the contrastive learning instruction tuning to enhance the model's confidence and avoid the situation where the model knows the knowledge but thinks it doesn't know. Another negative instance is a given question with incorrect answers generated by the LLM, applied to help the LLM forget the incorrect knowledge during training. It is worth noting that the selection of instances is related to the specific parameters of our threshold. When the $IK$ is chosen as 1.0 for the upper threshold, it means that the question will only be treated as a known question if the model answers all of them correctly during all sampling instances. At this point in the construction of the negative instances, there will be no corresponding incorrect answer negative instance, because questions with incorrect answers are already categorized in questions that the model does not know.

### 3.2.2 QUESTIONS THAT THE MODEL DOES NOT KNOW

Questions that are answered with a correctness rate less than the upper threshold $IK$, including "the model does not know that it knows" and "the model does not know that it does not know", fall into the two quadrants on the bottom of the knowledge quadrants, and are therefore considered as "model doesn't know". The negative instance of the two kinds of questions is a given question with incorrect answers generated by the LLM. In the construction of the positive instances, for questions with an accuracy greater than the lower threshold $IDK$, we consider that the model actually has the relevant knowledge, but is just not sure whether it knows it or not, which belongs to the knowledge "the model does not know that it knows". We set the questions with correct responses as positive instances, hoping to enhance the model's mastery of uncertain knowledge by introducing such knowledge.

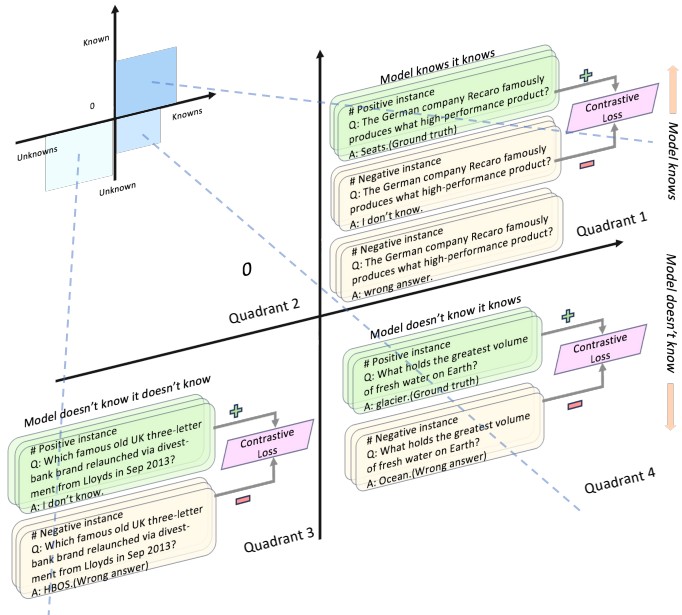

Figure 3: The examples of the contrastive learning data. $Quadrant - 2$ should contain the knowledge that models know that they don't know. Unfortunately, the initial model without our optimization does not have the ability to distinguish this kind of knowledge.

For questions with a correct response rate less than the lower threshold $IDK$, we consider that the model does not have the relevant knowledge, which belongs to the knowledge that "the model does not know that it does not know". We design the corresponding question and the "I don't know" answer as a positive instance. By narrowing the distance between this positive sample, we can better encourage the model to admit "I don't know" when facing the knowledge that the model doesn't have and enhance the honesty and creditability of the LLM responses.

To compute the adaptive contrastive loss in Section 3.3, we need to construct different contrastive data for various quadrants. Specifically, we construct different instruction data $I = (x, y)$, where $x$ is the input question and $y$ is the answer. In the contrastive setup, besides the anchor $I$, there are positive instances $I^+$ and negative instances $I^-$. The differences primarily focus on the selection of $y$, which will be detailed in the following sections.

## 3.3 ADAPTIVE CONTRASTIVE LEARNING STRATEGY

In Section 3.2, we obtain the constructed instruction pairs for correlated positive and negative instances of model-knowns versus model-unknowns knowledge from the three quadrants. We then perform adaptive contrastive instruction tuning. Compared to the traditional SFT, the contrastive instruction tuning approach can better enhance the model's mastery of knowledge by pulling the positive instances closer together and pushing the negative instances away Yan et al. (2024). In our adaptive contrastive learning strategy, for the positive and negative instances corresponding to knowledge in different quadrants, we design different data using strategy, which can adaptively perform the calculation of positive and negative samples in different quadrants. So that the model can be more targeted to think about different knowledge, and better enhance the model's ability to maintain what it knows, consolidate what it knows but is uncertain about, and forget unknown knowledge.

For the knowledge that the "models know that they know" in $Quadrant - 1$ (as shown in Figure 3), we have the contrastive data $I_{Q_1}$, $I_{Q_1}^+$, and $I_{Q_1}^-$. The $i$-th sample in a training batch includes the original input $I_{Q_1}^i$ (its $y^i$ is the golden answer), positive instance $I_{Q_1}^{i,+}$ (its $y^i$ is the previous correct answer of LLM), and negative instance $I_{Q_1}^{i,-}$ (its $y^i$ is "I don't know" or previous wrong answers of LLM). We bring the positive instance $I_{Q_1}^{i,+}$ closer and push the negative instance $I_{Q_1}^{i,-}$ farther away.

To calculate the contrastive loss, we obtain the hidden states $\mathbf{h}$ of $I$ following the method of Yan et al. (2024) and have designed a contrastive loss function:

$$\mathcal{L}_{Q_1}^{i,ctr} = -\log \frac{e^{\cos(\mathbf{h}_{Q_1}^i, \mathbf{h}_{Q_1}^{i,+})/\tau}}{e^{\cos(\mathbf{h}_{Q_1}^i, \mathbf{h}_{Q_1}^{i,+})/\tau} + e^{\cos(\mathbf{h}_{Q_1}^i, \mathbf{h}_{Q_1}^{i,-})/\tau}}, \tag{1}$$

where $\cos(\mathbf{h}_{Q_1}^i, \mathbf{h}_{Q_1}^{i,+})$ is the cosine similarity, and $\tau$ is a temperature hyperparameter.

In order to maintain the generative capability of the LLM, we follow the method of Liu et al. (2022) and design the standard cross-entropy loss for each instruction pair:

$$\mathcal{L}_{Q_1}^{i,gen} = \frac{1}{l} \sum_{k=1}^{l} -\log p(y_k^i | I_{Q_1}^i, y_{<k}^i), \tag{2}$$

where $l$ is the length of the desired output for the input, $y_k^i$ is the $k$-th token of $y^i$, $y_{<k}^i$ is the previous tokens before $y_k^i$.

The overall learning objective function was obtained by combining the above two loss functions:

$$\mathcal{L}_{Q_1}^{i,Adap} = \mathcal{L}_{Q_1}^{i,gen} + \max(\lambda, \text{detach}(\frac{\mathcal{L}_{Q_1}^{i,gen}}{\mathcal{L}_{Q_1}^{i,ctr}})) \mathcal{L}_{Q_1}^{i,ctr}, \tag{3}$$

where detach denotes that the loss value is detached from the computation graph and thus is treated only as a scalar, and $\lambda$ is the upper bound of the weight.

For the knowledge that the model does not know that it does not know in $Quadrant-3$ and the model does not know that it knows in $Quadrant-4$ (as shown in Figure 3), we design similar objective functions, as described in detail in the Appendix. A.1.

## 4 EXPERIMENT

### 4.1 EXPERIMENTAL SETUP

#### 4.1.1 TRAINING DETAILS OF METHODS

In order to verify the superiority of our proposed contrastive learning fine-tuning strategy, we design different LLM-based methods as baselines, and implement the baselines with the same details for fair comparison. Their training details are presented here.

**IDK Prompting.** In order to equip the model with the initial honesty of refusing to answer questions that it does not know, we designed the Prompting scheme directly for the baseline large model. In the Prompt, the model is told to reply "I don't know" when it encounters a question that it cannot answer. This approach requires the model not only to have good command following ability, which can correctly follow the instructions provided by us, but also to have the ability to distinguish its own knowledge boundaries, i.e., to distinguish what is a question that it cannot answer. In IDK Prompting, the model is not additionally trained or fine-tuned, and the model's own ability to follow instructions and distinguish between knowledge boundaries is completely examined.

**IDK SFT.** Supervised fine-tuning (SFT) is a very important technique for fine-tuning LLMs and can be very useful to help models quickly learn new knowledge patterns from constructed high-quality datasets. We fine-tune the model based on the problems that the model knows versus the problems that the model does not know, from the contrastive learning data we construct. In the input data, each pair of instances consists of a question and a response, where the questions that the model knows correspond to correct responses and the questions that the model does not know correspond to "I don't know" responses. By calculating the standard sequence-to-sequence loss of the model-generated responses with respect to the standard responses, we train the model.

#### 4.1.2 IMPLEMENTATION DETAILS

We select LLaMA-2-7B-chat and Mistral-7B-Instruct-v0.1 as base models for testing on TriviaQA (Joshi et al., 2017) and Natural Questions (Kwiatkowski et al., 2019), respectively. During the

| Method | LLaMA-2-7B-chat | | | | | |
| | TriviaQA | | | Natural Questions | | |
| | IK-IK | IK-IDK | TRUTHFUL | IK-IK | IK-IDK | TRUTHFUL |
|---|---|---|---|---|---|---|
| IDK-Prompting | 37.4 | 29.6 | 66.9 | 19.7 | 41.4 | 61.1 |
| IDK-SFT | 28.0 | 45.2 | 73.2 | 32.8 | 27.5 | 60.3 |
| IDK-SFT-Adpt-Ctr | 37.3 | 40.9 | 78.2 | 21.8 | 45.4 | 67.2 |
| Method | Mistral-7B-Instruct-v0.1 | | | | | |
| | TriviaQA | | | Natural Questions | | |
| | IK-IK | IK-IDK | TRUTHFUL | IK-IK | IK-IDK | TRUTHFUL |
| IDK-Prompting | 50.8 | 5.9 | 56.7 | 23.6 | 20.8 | 44.4 |
| IDK-SFT | 31.4 | 40.8 | 72.2 | 4.8 | 65.4 | 70.2 |
| IDK-SFT-Adpt-Ctr | 24.4 | 49.0 | 73.5 | 9.3 | 67.7 | 77.0 |

Table 1: Overall results on the test set of the IDK dataset constructed based on TriviaQA and out-of-distribution test sets.

training of the LLaMA model, we used a batch size of 16, a learning rate of 5e-5, a context length of 1024, and trained for 2 epochs. For the Mistral model, we used a batch size of 16, a learning rate of 1e-5, a context length of 1024, and also trained for 2 epochs. The $\tau$ is set to 0.01 and the $\lambda$ is set to 1. All experiments are conducted on Nvidia A100 80GB GPUs. During inference, we utilize the vllm framework to accelerate the process and employ a greedy search strategy to generate responses.

## 4.2 DATASETS

**TriviaQA** (Joshi et al., 2017) is a reading comprehension dataset with question-answering pairs widely used in open-domain quizzing, which contains 87,622 pairs in the training set. Referring to past work Cheng et al. (2024), we use 90% of the training set to construct a training set for comparative learning data and 10% as a validation set. Since there is no standard answer in TriviaQA's test set, we select 11,313 Q&A pairs from the development set to build our final test set.

**Natural Questions** (Kwiatkowski et al., 2019) is fine-tuned by constructing a comparative learning dataset on TriviaQA, and we perform tests with the same data distribution. In order to validate the performance of our fine-tuning method on OOD data, we select Natural Questions as our test dataset. Natural questions are real Q&A pairs from the Google search engine, where the development set containing 3,610 instances is used to build our test set. Same as when building the data in TriviaQA, we also use lexical matching as a metric for automatic evaluation when testing on Natural Questions.

## 4.3 EVALUATION METRICS

We employ the following evaluation metrics: 1) **IK-IK Rate**: This metric reflects the model's "I know what I know" capability, which is calculated as the percentage of correct answers relative to the total number of questions. 2) **IK-IDK Rate**: This represents the model's "I know what I don't know" ability, which calculates the ratio of those questions that correctly refuse answering to the total number of questions. 3) **Truthful Rate**: Both correct answers and correct refusals to answer outside knowledge boundaries are considered reliable, so the Truthful Rate is computed as the sum of the IK-IK and IK-IDK rates, and regarded as a comprehensive measure of its knowledge refinement capabilities. We introduce the evaluation metrics in detail in Appendix B.1.

## 4.4 MAIN RESULTS

From the Table 1, we can observe that in the TriviaQA dataset using the LLaMA model, the model fine-tuned with SFT shows a 6.3% improvement in the Truthful Rate compared to directly using IDK-Prompting. After incorporating our Adaptive Loss, there is an additional 5.0% improvement over the SFT model. Notably, the IK-IK rate shows a significant increase of 9.3% compared to the SFT model, indicating that our Adaptive Loss helps mitigate the loss of knowledge that the model is unaware it possesses due to SFT. On the OOD dataset, Natural Question, our model achieves improvements of 6.1% and 6.9% over IDK-Prompting and IDK-SFT, respectively.

In the case of the Mistral model on the TriviaQA dataset, the model fine-tuned with SFT exhibits a 15.5% improvement in the Truthful Rate compared to directly using IDK-Prompting. Building on this significant improvement, the addition of Adaptive Loss further enhances performance by 1.3%. On the OOD dataset, Natural Question, our model achieves improvements of 15.9% and 6.8% over IDK-Prompting and IDK-SFT, respectively.

| Knowing Rate | LLaMA-2-7B-chat | | | | | |
| | TriviaQA | | | Natural Questions | | |
| | IK-IK | IK-IDK | TRUTHFUL | IK-IK | IK-IDK | TRUTHFUL |
|---|---|---|---|---|---|---|
| 0.5 | 42.7 | 34.5 | 77.2 | 29.8 | 32.4 | 62.2 |
| 0.7 | 37.3 | 40.9 | 78.2 | 21.8 | 45.4 | 67.2 |
| 0.9 | 15.7 | 57.9 | 73.6 | 13.5 | 50.3 | 63.8 |

Table 2: The impact of different $IDK$ values on the performance of LLaMA-2-7B-chat.

## 5 DISCUSSION

### 5.1 IMPACT OF DIFFERENT $IDK$ VALUES

In Section 3.1 and Section 3.2, we utilize the $IDK$ Rate threshold to distinguish between the model's uncertain knowledge (the model does not know that it knows) and unknown knowledge (the model does not know that it does not know). In this process, we aim to refine the model's correct knowledge and reduce its uncertainty regarding the former, while encouraging it to forget incorrect knowledge regarding the latter. We conducted experiments with different $IDK$ Rates to evaluate model performance under varying thresholds. As shown in Table 2, the model achieves optimal performance at an IDK Rate of 0.7, followed by 0.5, with 0.9 being the lowest. We believe that categorizing too many uncertain responses as "the model does not know that it knows" can harm performance, as forcing the model to assert knowledge in low-certainty situations can have negative effects. Conversely, a high certainty threshold of 0.9 may prevent the model from reaching its full potential.

| Loss Combination | TriviaQA | | |
| | IK-IK | IK-IDK | TRUTHFUL |
|---|---|---|---|
| $\mathcal{L}_{Q_1}$ (Model knows it knows) | 28.7 | 46.0 | 74.7 |
| $\mathcal{L}_{Q_3}$ (Model doesn't know it doesn't know) | 30.0 | 43.8 | 73.8 |
| $\mathcal{L}_{Q_4}$ (Model doesn't know it knows) | 36.5 | 39.3 | 75.8 |
| $\mathcal{L}_{Q_1} + \mathcal{L}_{Q_3}$ | 26.5 | 47.9 | 74.4 |
| $\mathcal{L}_{Q_1} + \mathcal{L}_{Q_4}$ | 32.9 | 45.1 | 78.0 |
| $\mathcal{L}_{Q_3} + \mathcal{L}_{Q_4}$ | 29.4 | 46.7 | 76.1 |
| Total | 37.3 | 40.9 | 78.2 |

Table 3: Results of different loss combinations. Experiments are conducted on LLaMA-2-7B-chat.

### 5.2 IMPACT OF DIFFERENT CONTRASTIVE LOSS COMBINATIONS

In designing the Adaptive Contrastive Learning strategy, we developed different contrastive learning strategies based on the model's certainty regarding various issues. Here, we isolate each type of contrastive learning loss by setting the others to zero to evaluate the impact of each specific loss type. From Table 3, we can see that:

- With the $\mathcal{L}_{Q_1}$ (i.e., "Model knows it knows" in $Quadrant-1$), the model's responses are more conservative, resulting in a lower IK-IK rate and a higher IK-IDK rate.

- The $\mathcal{L}_{Q_3}$ (i.e., "Model doesn't know it doesn't know" in $Quadrant-3$) encourages the model to be conservative and forget incorrect knowledge, resulting in a lower IK-IK rate.

- Using the $\mathcal{L}_{Q_4}$ (i.e., "Model doesn't know it knows" in $Quadrant-4$) allows the model to correctly answer more uncertain questions, leading to a higher IK-IK rate.

- For $\mathcal{L}_{Q_1} + \mathcal{L}_{Q_3}$, the model has higher IK-IDK rate due to cautious responses, but lower IK-IK than Total. Missing $\mathcal{L}_{Q_4}$ limits the use of underlying knowledge for confident answers.

- With $\mathcal{L}_{Q_1} + \mathcal{L}_{Q_4}$, the model sees strong Truthful rate and decent IK-IK results. However, without $\mathcal{L}_{Q_3}$, it's less cautious about unknowns, affecting the IK-IDK rate.

- The $\mathcal{L}_{Q_3} + \mathcal{L}_{Q_4}$ provides balanced performance but lower IK-IK rate comparing Total. This suggests that without $\mathcal{L}_{Q_1}$, the model struggles with certainty in known answers.

Combining these three components forms our method, which achieves superior overall performance.

### 5.3 REPEATED SAMPLING DISTRIBUTION

To more intuitively demonstrate the improvements of our method over IDK-SFT, we conduct ten repeated samplings of questions from the TriviaQA test set on LLaMA-2-7B-chat, categorizing them into "Unknown Questions" and "Known Questions." When calculating accuracy, the ground truth for "Known Questions" is the correct answer, while for "Unknown Questions", it is a refusal to respond. The distribution of our sampling results is shown in Figure 4. In this figure, the vertical axis represents the proportion of samples within the entire dataset that fall into a specific accuracy range. We employ repeated sampling to evaluate both models by having them respond to each question in the TriviaQA test set 10 times. The number of correct responses out of these 10 attempts (i.e., accuracy) is used as a measure of the model's performance and confidence. To explain Figure 4, in the "Unknown Questions" sub-figure of Figure 4, the bars corresponding to an accuracy of 1 indicate the proportion of questions from the dataset that the models consistently declined to answer across all 10 attempts. Similarly, in the "Known Questions" sub-figure, the bars corresponding to an accuracy of 0.8 represent the proportion of questions for which the models provided 8 correct answers out of 10 attempts, relative to the total number of questions in the dataset.

From Figure 4, we observe that our method achieves a higher number of instances with high accuracy (greater than 0.7) in both Unknown and Known questions compared to Idk-SFT. This indicates that the model fine-tuned with our method is more likely to produce correct results in a single response.

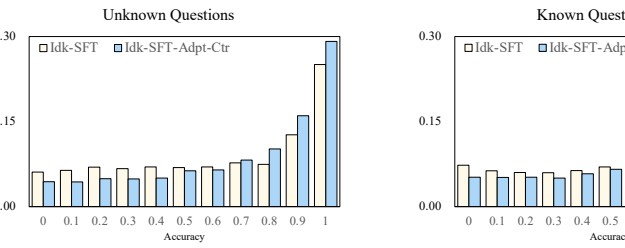

Figure 4: Accuracy Distribution for Known and Unknown Questions in TriviaQA.

### 6 CONCLUSION

In this paper, we focus on the problem of hallucination in LLM responses. We believe that this hallucination arises from the LLM's choice to reply despite being confronted with a question that exceeds its own knowledge. This motivates our exploration of the LLM's mastery of its own internal knowledge and its knowledge boundaries. We design a new knowledge boundary delineation method, which helps the model better represent and refine its own knowledge quadrants. We propose the strategy of Adaptive Contrastive Learning, by targeting different knowledge mastery abilities in different knowledge quadrants, we design different positive and negative samples with adaptive loss functions to help the model maintain known knowledge, consolidate uncertain knowledge, and forget the wrong knowledge. We conduct experiments on in-distribution and out-of-distribution datasets, and the results show that this proposed contrastive learning strategy well improves the Truthful rate of the models. We further provide presentations on threshold analysis, loss function ablation experiments, and visualization of results with knowledge quadrants, which not only further demonstrates the validity of our results, but also provides valuable inspiration for future work.

ACKNOWLEDGEMENTS

This research is supported by the National Natural Science Foundation of China (Grant No. 62276154), the Natural Science Foundation of Guangdong Province (Grant No. 2023A1515012914 and 44030024103310080177O), Shenzhen Science and Technology Program (Grant No. WDZC20231128091437002), Basic Research Fund of Shenzhen City (Grant No. JCYJ20210324120012033, GJHZ202402183000101, and JCYJ20240813112009013), the Major Key Project of PCL for Experiments and Applications (PCL2022A05 and PCL2023A09). This work is also supported in part by NSF under grants III-2106758, and POSE-2346158.

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

## A    ADDITIONAL DETAILS OF METHODOLOGY

### A.1    OBJECTIVE FUNCTION

**The model does not know it knows.** For the knowledge that the "model doesn't know that it knows" in $Quadrant - 4$ (as shown in Figure 3), we have the contrastive data $I_{Q_4}$, $I_{Q_4}^+$, and $I_{Q_4}^-$. The $i$-th sample in a training batch includes the original input $I_{Q_4}^i$ (its $y^i$ is the golden answer), positive instance $I_{Q_4}^{i,+}$ (its $y^i$ is the golden answer), and negative instance $I_{Q_4}^{i,-}$ (its $y^i$ is previous incorrect answers of LLM). We bring the positive instance $I_{Q_4}^{i,+}$ closer and push the negative instance $I_{Q_4}^{i,-}$ farther away. We design the loss function:

$$\mathcal{L}_{Q_4}^{i,ctr} = -\log \frac{e^{\cos(\mathbf{h}_{Q_4}^i, \mathbf{h}_{Q_4}^{i,+})/\tau}}{e^{\cos(\mathbf{h}_{Q_4}^i, \mathbf{h}_{Q_4}^{i,+})/\tau} + e^{\cos(\mathbf{h}_{Q_4}^i, \mathbf{h}_{Q_4}^{i,-})/\tau}}, \tag{4}$$

where $\cos(\mathbf{h}_{Q_4}^i, \mathbf{h}_{Q_4}^{i,+})$ is the cosine similarity, and $\tau$ is a temperature hyperparameter.

Again, there is cross-entropy loss with the final objective function:

$$\mathcal{L}_{Q_4}^{i,gen} = \frac{1}{l} \sum_{k=1}^{l} -\log p(y_k^i | I_{Q_4}^i, y_{<k}^i), \tag{5}$$

$$\mathcal{L}_{Q_4}^{i,Adap} = \mathcal{L}_{Q_4}^{i,gen} + \max(\lambda, \text{detach}(\frac{\mathcal{L}_{Q_4}^{i,gen}}{\mathcal{L}_{Q_4}^{i,ctr}}))\mathcal{L}_{Q_4}^{i,ctr}, \tag{6}$$

where detach denotes that the loss value is detached from the computation graph and thus is treated only as a scalar, and $\lambda$ is the upper bound of the weight.

**The model does not know that it does not know.** For the knowledge that the "models do not know that they do not know" in $Quadrant - 3$ (as shown in Figure 3), we have the contrastive data $I_{Q_3}$, $I_{Q_3}^+$, and $I_{Q_3}^-$. The $i$-th sample in a training batch includes the original input $I_{Q_3}^i$ (its $y^i$ is "I don't know"), positive instance $I_{Q_3}^{i,+}$ (its $y^i$ is "I don't know"), and negative instance $I_{Q_3}^{i,-}$ (its $y^i$ is previous incorrect answers of LLM). We bring the positive instance $I_{Q_3}^{i,+}$ closer and push the negative instance $I_{Q_3}^{i,-}$ farther away. We design the loss function:

$$\mathcal{L}_{Q_3}^{i,ctr} = -\log \frac{e^{\cos(\mathbf{h}_{Q_3}^i, \mathbf{h}_{Q_3}^{i,+})/\tau}}{e^{\cos(\mathbf{h}_{Q_3}^i, \mathbf{h}_{Q_3}^{i,+})/\tau} + e^{\cos(\mathbf{h}_{Q_3}^i, \mathbf{h}_{Q_3}^{i,-})/\tau}}, \tag{7}$$

## B    ADDITIONAL DETAILS OF EXPERIMENT

### B.1    EVALUATION METRICS

We use the following evaluation metrics, all of the evaluation scores show that the higher the score, the better the model effect:

**IK-IK rate:**    This means that the model "I know what I know", which means the model does not refuse to answer the question, but gives a correct answer, and the question is labeled as "model knows". We calculate the number of correct answers given by the model as a percentage of the total number of questions in the dataset as a fraction of the IK-IDK rate.

**IK-IDK rate:**    This represents the model's "I know what I don't know", meaning that the model refuses to answer a question that is labeled as a "model-don't-know question" in the test set. When we detect the presence of our predetermined "I don't know", we calculate the ratio of the number of questions that the model refuses to answer to the total number of questions as the IK-IDK rate.

**Truthful Rate:** This represents the overall percentage of reliable answers given by the model. In the use of large models, for the model giving correct responses, versus refusing to respond to responses that are outside the knowledge boundaries, both responses can be considered reliable, so the Truthful rate is calculated as the sum of IK-IK and IK-IDK. Since all data are categorized into "model knowns" and "model unknowns", although the upper bounds for the separate scores for IK-IK and IK-IDK may vary depending on the data categorization, ideally the Truthful Rate can reach 100. Therefore, we believe that the Truthful Rate is a more comprehensive criterion for assessing the intellectual honesty of a model, as the upper bound is not affected by data partitioning, and it is a more comprehensive measure of the model's mastery of the capability to refine knowledge.

## C ADDITIONAL EXPERIMENT

This section is entirely new, so highlighting is omitted.

### C.1 MORE DATASETS

**Experimental Setup** To evaluate the model's ability to respond appropriately to completely unknown questions, we selected the ALCUNA(Yin et al., 2023a) dataset for testing. This dataset is designed by creating artificial entities through the alteration of existing entity attributes, leading to the generation of questions about these novel entities. Since these entities are artificially constructed, it is nearly impossible for the model to possess prior knowledge about them, making this dataset ideal for testing the model's ability to refuse to answer inconceivable queries. We randomly sampled 1000 instances from the ALCUNA dataset to serve as our out-of-domain test set.

| Method | ALCUNA | | |
|---|---|---|---|
| | IK-IK | IK-IDK | TRUTHFUL |
| IDK-Prompting | 1.6 | 90.3 | 91.9 |
| IDK-SFT | 1.2 | 96.6 | 97.8 |
| IDK-SFT-Adpt-Ctr | 1.0 | 97.3 | 98.3 |

Table 4: Performance of different methods on the ALCUNA dataset.

**Results Analysis** As shown in Table 4, our proposed methods perform better in the context of unknown queries that require refusal to respond, compared to models that have only finetuned by IDK-SFT. This underlines the effectiveness of our approach in reinforcing the model's ability to manage queries about unknown entities.

| Method | TriviaQA | | |
|---|---|---|---|
| | IK-IK | IK-IDK | TRUTHFUL |
| IDK-Prompting | | | |
| - with *LLaMA-2-7B-chat* | 37.4 | 29.6 | 66.9 |
| - with *LLaMA-2-13B-chat* | 37.7 | 31.6 | 69.3 |
| IDK-SFT | | | |
| - with *LLaMA-2-7B-chat* | 28.0 | 45.2 | 73.2 |
| - with *LLaMA-2-13B-chat* | 32.8 | 41.3 | 74.1 |
| IDK-SFT-Adpt-Ctr | | | |
| - with *LLaMA-2-7B-chat* | 37.3 | 40.9 | 78.2 |
| - with *LLaMA-2-13B-chat* | 37.8 | 41.1 | 78.9 |

Table 5: Performance of different model sizes on the TriviaQA dataset.

## C.2 IMPACT OF MODEL SIZE ON PERFORMANCE

**Experimental Setup**    To investigate the effect of different model sizes on our proposed method, we conducted experiments using the LLaMA-2-13B-chat model. The experimental setup was consistent with the previous sections. By comparing with the LLaMA-2-7B-chat model, we aimed to understand how scaling the model impacts performance across various metrics.

**Results Analysis**    The results, as presented in Table 5, demonstrate nuanced changes in performance as the model size increases from 7B to 13B parameters. For all method, using the larger LLaMA-2-13B-chat model resulted in slight improvements across IK-IK, IK-IDK, and Truthful rate, indicating that a larger model can enhance the ability to differentiate known and unknown knowledge and present truthful responses. Moreover, the proposed method consistently gained performance improvements across all metrics with the larger LLaMA-2-13B-chat model, thereby underscoring the robustness and scalability of our approach.

## C.3 COMPARISON WITH RAG INTEGRATION

**Experimental Setup**    To assess the adaptability of our method, we integrated the Retrieval-Augmented Generation (RAG) technique. We conduct experiments using the LLaMA-2-7B-chat model, and we employ the **RAG-Bench** (Fang et al., 2024) dataset, constructed from three open-domain question answering datasets: NQ (Cheng et al., 2024), TriviaQA (Joshi et al., 2017), and WebQ (Berant et al., 2013). For each dataset, a retrieval model sourced relevant paragraphs from Wikipedia for each query to use as context. In this experimental setup, we input the context alongside the query into the model (denoted as "with RAG" in the table) and compared the results with those obtained without using context. We randomly selected 1000 samples from RAG-Bench to use as our out-of-domain test test.

| Method | RAG-Bench | | |
|---|---|---|---|
| | IK-IK | IK-IDK | TRUTHFUL |
| IDK-Prompting | 47.4 | 11.3 | 58.7 |
| - with RAG | 62.5 | 5.8 | 68.3 |
| IDK-SFT | 60.3 | 1.9 | 62.2 |
| - with RAG | **66.1** | 3.3 | 69.4 |
| IDK-SFT-Adpt-Ctr | 44.5 | **22.4** | 66.9 |
| - with RAG | 58.4 | 12.8 | **71.2** |

Table 6:   Performance comparison using LLaMA-2-7B-chat on the RAG-Bench dataset with and without RAG integration.

**Results Analysis**    As illustrated in Table 6, integrating the RAG technique positively impacted the Truthful Rate of all three methods. Additionally, it can be seen that IDK-Prompting and IDK-SFT-Adpt-Ctr show a decreasing trend in IK-IDK after combining with RAG, while IDK-SFT shows an increasing trend in IK-IDK after combining with RAG. The reasons for this phenomenon are as follows:

1. It is intuitive and reasonable that IDK-Prompting and IDK-SFT-Adpt-Ctr exhibit a decreasing trend in IK-IDK when combined with RAG, as the model naturally tends to answer questions more directly and reduce refusal-to-answer after being input with more additional contextual information from RAG. Therefore, from the perspective of metrics, after adding RAG to these two methods, IK-IDK decreased.

2. The phenomenon of the increase in IK-IDK after adding RAG to IDK-SFT can only indicate an increase in correct refusal-to-answer, but this does not conflict with the above analysis, as the overall refusal-to-answer include both correct and wrong ones. After careful observation, we found that the overall rejection rate of IDK-SFT is about 10%, while the overall rejection rate of IDK-SFT-RAG is about 4%. This indicates that the overall rejection rate of IDK-SFT still decreases after adding RAG, and the main reason for the increase in IK-IDK is that a considerable portion of refusal-to-answer in IDK-SFT is wrong.

