# OpenReview forum: "Refine Knowledge of Large Language Models via Adaptive Contrastive Learning"
_ICLR.cc/2025/Conference — ICLR 2025 Poster_

### Official Review · Reviewer_ebFf · 2024-10-29

**Soundness:** 3
**Presentation:** 4
**Contribution:** 3
**Rating:** 6
**Confidence:** 4

**Summary:**

In this paper, the authors emulate the human learning process by proposing an **Adaptive Contrastive Learning** strategy. This approach dynamically constructs positive and negative samples based on the actual mastery of knowledge in Large Language Models (LLMs). It enhances LLM's understanding of the correct knowledge they have encountered but not fully grasped, allows them to discard incorrect knowledge they previously learned, and candidly recognizes the knowledge they lack. Extensive experiments and detailed analyses demonstrate the effectiveness of this method.

**Strengths:**

- **Innovative Knowledge Representation Method:** This is a well-written paper. The knowledge quadrant refines the boundaries of an LLM’s knowledge representation, aligning it more closely with real-world human knowledge needs.

- **Well-Motivated Adaptive Contrastive Learning Strategy:** By maximizing the distance between negative samples and minimizing the distance between positive samples, the model learns to preserve its existing knowledge, consolidate known but uncertain information, and forget unknown knowledge. This approach enhances the validity and honesty of the model’s responses.

- **Effective Approach:** Experimental results demonstrate that the strategy achieves the highest truthful rate across various advanced LLMs with both in-distribution and out-of-distribution data, thereby confirming the effectiveness of the proposed strategy.

**Weaknesses:**

- In this work, the discussion surrounding the concept of "I know that I don't know" (Quadrant 2) is lacking. Could you provide an explanation of how to differentiate between the knowledge categories of "I know that I know" and "I know that I don't know"? Specifically, would you categorize information as "I know that I don't know" if the model consistently responds with "I don't know" across all sampling responses?

**Questions:**

- How does varying the inference temperature affect the model confidence in your setting?

---

> ### Author Response · Authors · 2024-11-20
> **Response to reviewer ebFf**
>
> We greatly appreciate the valuable feedback from the reviewer, and hope our response can address your concerns.
>
> > Weakness: In this work, the discussion surrounding the concept of "I know that I don't know" (Quadrant 2) is lacking.
>
> We apologize for the confusion with our description of “I know that I don't know”. In our work, we believe that when LLM gives a correct answer to a question as a response, it is regarded as “I know that I know”. When sampling responses, a question is considered as “I know that I don't know” when the model not only responds “I don't know”, but also satisfies the condition that it is really beyond the model's knowledge. In fact, it is possible for a model to respond “I don't know” to a question for which it knows the correct answer. At this point, we consider the model “I don't know that I know”. Whether or not the model actually understands the answer to the question is determined by the response accuracy of our sampling. It is worth noting that, as we sample responses, we observe that without additional prompts or fine-tuning, the LLM tends to give a response even if it doesn't know the correct answer, so it almost never chooses to reject a response.
>
> In our experiments, we added an additional prompt that told LLM to reply “I don't know” to questions it did not know. At this point, we regard those questions with the response “I don't know” that are labeled as questions for which the model really did not know the corresponding knowledge in our data construction as the model's “I know that I don't know”.
>
> > Question: How does varying the inference temperature affect the model confidence in your setting?
>
> Thank you very much for your suggestion. We refer to past work [1][2] and note that they do not explore different temperatures, but rather use constant tempareture for reasoning. **Therefore, for the sake of a fair comparison, we choose the same temperature value of 0.7 as theirs, without additional modifications and further exploration.**
>
> [1] [R-Tuning: Instructing Large Language Models to Say ‘I Don’t Know’](https://aclanthology.org/2024.naacl-long.394) (Zhang et al., NAACL 2024)
>
> [2] [Can AI Assistants Know What They Don't Know?](https://proceedings.mlr.press/v235/cheng24i.html) (Cheng et al., ICML 2024)

---

> > ### Comment · Reviewer_ebFf · 2024-11-25
> > **Thank you for your explanation!**
> >
> > I thank the authors for the thorough explanation. Their response has addressed my concern.

---

### Official Review · Reviewer_5ff6 · 2024-11-03

**Soundness:** 2
**Presentation:** 2
**Contribution:** 2
**Rating:** 6
**Confidence:** 3

**Summary:**

This paper primarily addresses the issue of hallucination in large language models (LLMs), particularly when models provide incorrect or fabricated answers to questions they cannot answer, severely impacting model trustworthiness.

**Strengths:**

The paper is well-structured, especially in its quadrant-based classification of knowledge and its innovative use of an adaptive contrastive learning strategy, demonstrating a rigorous design approach. By combining knowledge classification with an adaptive loss function, it presents a novel solution to effectively address the hallucination problem in LLMs.

**Weaknesses:**

First, the experiment coverage is relatively narrow, with results demonstrated on only two datasets. Although the method shows promising results, its broad applicability across more tasks and domains is uncertain. Expanding the evaluation to more diverse datasets, such as dialogue or fact-checking datasets, would help validate the robustness and applicability of the method.
Second, although the paper proposes using a combination of three loss functions for different types of knowledge, it does not delve into the interactions and balance between these loss functions. In different scenarios, each loss may contribute differently; exploring the interactions and balance of these loss functions across various tasks would further strengthen the theoretical foundation of this method.

**Questions:**

I don't have questions.

---

> ### Author Response · Authors · 2024-11-20
> **Response to reviewer 5ff6**
>
> We greatly appreciate the valuable feedback from the reviewer. To address the reviewer's concerns, we have conducted additional experiments:
>
> > Weakness 1: First, the experiment coverage is relatively narrow, with results demonstrated on only two datasets. Although the method shows promising results, its broad applicability across more tasks and domains is uncertain. Expanding the evaluation to more diverse datasets, such as dialogue or fact-checking datasets, would help validate the robustness and applicability of the method.
>
> **Response**
>
> In addition to the two experiments mentioned in the original text, **we have conducted an additional experiment using the LLaMA-2-7B-chat model on the Alcuna dataset.** This dataset is designed by creating artificial entities through the alteration of existing entity attributes, leading to the generation of questions about these novel entities. Since these entities are artificially constructed, it is nearly impossible for the model to possess prior knowledge about them, making this dataset ideal for testing the model’s ability to refuse to answer inconceivable queries. We conduct this experiement to evaluate the model's ability to respond appropriately to completely unknown questions.
>
> |                   | ALCUNA      |       |          |
> |-------------------|-----------------------|-------|----------|
> |                   | IK-IK                 | IK-IDK| TRUTHFUL |
> | IDK-Prompting     | 1.6                   | 90.3  | 91.9     |
> | IDK-SFT           | 1.2                   | 96.6  | 97.8     |
> | IDK-SFT-Adpt-Ctr       | 1.0                   | 97.3  | 98.3     |
>
> As we can see in the table, our proposed methods perform better in the context of unknown queries comparing with the models that finetuned with IDK-SFT. The results show that the proposed method effectively avoid giving misleading answers, proving reliability when dealing with unknown areas. **These findings have be updated in Appendix C.1 of our revised paper.**
>
> > Weakness 2: Second, although the paper proposes using a combination of three loss functions for different types of knowledge, it does not delve into the interactions and balance between these loss functions. In different scenarios, each loss may contribute differently; exploring the interactions and balance of these loss functions across various tasks would further strengthen the theoretical foundation of this method.
>
> **Response**
>
> We sincerely appreciate the reviewer's insightful comments regarding the interaction and balance between the loss functions. **In response to your feedback, we have designed experiments to explore these interactions. Specifically, we utilized our Adaptive Contrastive Learning strategy to test different combinations of our loss functions.** We paired the three loss functions in two's to observe their interactions. The results of these experiments are presented in the following table:
>
> | Loss Combination                             | TriviaQA      |       |          |
> |----------------------------------------|---------------|-------|----------|
> |         | IK-IK         | IK-IDK| TRUTHFUL |
> | $L_{Q_1}$          | 28.7        | 46.0 | 74.7    |
> | $L_{Q_3}$         | 30.0   | 43.8 | 73.8    |
> | $L_{Q_4}$          | 36.5   | 39.3 | 75.8    |
> | $L_{Q_1}$ + $L_{Q_3}$   | 26.5 | 47.9 | 74.4 |
> | $L_{Q_1}$ + $L_{Q_4}$ | 32.9 | 45.1 | 78.0 |
> | $L_{Q_3}$ + $L_{Q_4}$ | 29.4 | 46.7 | 76.1 |
> |        Total         |   37.3| 40.9 | 78.2 |
>
> In this table, $L_{Q_1}$ indicates "Model knows it knows'' in $Quadrant-1$,  $L_{Q_3}$ indicates "Model doesn't know it doesn't know'' in $Quadrant-3$, and $L_{Q_4}$ indicates "Model doesn't know it knows'' in $Quadrant-4$. For $L_{Q_1} + L_{Q_3}$, the model has a higher IK-IDK rate due to cautious responses, but lower IK-IK compared to Total. Missing $L_{Q_4}$ limits the use of underlying knowledge for confident answers. With $L_{Q_1} + L_{Q_4}$, the model sees strong Truthful rate and decent IK-IK results. However, without $L_{Q_3}$, it's less cautious about unknowns, affecting the IK-IDK score. The $L_{Q_3} + L_{Q_4}$ combination provides balanced performance but lacks the strong IK-IK scores of Total. This suggests that without $L_{Q_1}$, the model struggles with certainty in known answers. **These findings have be updated in Section 5.2 of our revised paper.**
>
> Reference:
> [1][ALCUNA: Large Language Models Meet New Knowledge](https://aclanthology.org/2023.emnlp-main.87) (Yin et al., EMNLP 2023)

---

### Official Review · Reviewer_BPiT · 2024-11-03

**Soundness:** 3
**Presentation:** 2
**Contribution:** 3
**Rating:** 6
**Confidence:** 4

**Summary:**

This paper tries to mitigate the hallucination problem of large language models by proposing an Adaptive Contrastive Learning strategy. The method curates data that represent the zones of "model knows it knows", "model doesn't know it knows", "model doesn't know it doesn't know", and designs contrastive learning loss to strengthen the model's desired behavior to be more trustworthy when answering questions. Experimental results show that the proposed method outperforms "I don't know" prompting and "I don't know" SFT.

**Strengths:**

1. The results of the paper are solid, as the proposed method outperforms IDK-Prompting and IDK-SFT by a large margin. Contrastive learning is not a new method, but the paper provides a good example of how to use it to improve large language models with careful contrastive pairs and loss design.
2. The paper is well-motivated with well-made figures illustrating the data construction idea. The hallucination issue is a major block of using LLMs in high-stake scenarios, and many papers have thought of tuning the models to output "I don't know"[1]. However, the existing papers usually only consider the dichotomy of answerable and unanswerable questions without having a 2-D division of the space. It's interesting to see this paper brings out this concept as it's not only useful for helping model learning but could also help human learning [2].

[1] R-Tuning: Instructing Large Language Models to Say "I Don't Know", Zhang et al., NAACL 2024.

[2] Into the Unknown Unknowns: Engaged Human Learning through Participation in Language Model Agent Conversations, Jiang et al., EMNLP 2024.

**Weaknesses:**

1. Even though the results look strong by looking at the Truthful Rate, I am not sure whether the proposed Truthful Rate is well-defined. Given the denominators of IK-IK Rate and IK-IDK Rate are different, I don't think it's rigorous to add them together as the final score. An alternative could be treating answering "I don't know" to those answerable questions as correct and computing the overall accuracy on the full dataset. Also, why the IK-IK rate for Mistral-7B-Instruct-v0.1 drops drastically compared to IDK-prompting? This is a bit concerning as it indicates after Adaptive Contrastive Learning, the model may not be useful anymore.
2. i am unsure what's the major takeaway of Section 5.3. Does the accuracy under repeated samplings represent the concept of "confidence"? Actually, the paper mentions the term of "confidence" when introducing the motivation of the proposed method. It would be interesting to see whether the proposed method can draw a more rigorous connection of model's confidence in its generation.
3. The presentation for the paper can be further improved. See my detailed comments below.


I am happy to increase my score if these issues can be addressed.

**Update: I updated my score to indicate support of acceptance since many points were addressed by the revised version.**

**Questions:**

I think the paper can benefit from a careful round of proofreading. Here are some nits I discovered when reviewing the paper:
1. In Table 1, it's better to use "IDK" instead of "idk" to be accord with the main paper.
2. I suggest rewriting Section 5.2 as the current writing is a bit confusing, even though I can get the meaning by reading Table 3. For example, for the first bullet point, "With" should be "Without" based on my understanding.
3. I am wondering if the notations of the paper can be improved. For example, using $IDKIK$ as subscript in equations looks weird.

---

> ### Author Response · Authors · 2024-11-20
> **Response to reviewer BPiT (part 1)**
>
> Thank you very much for your recognition of our work and your insightful comments. We hope to address your concerns through our response:
>
> > Weakness 1:  Even though the results look strong by looking at the Truthful Rate, I am not sure whether the proposed Truthful Rate is well-defined. Given the denominators of IK-IK Rate and IK-IDK Rate are different, I don't think it's rigorous to add them together as the final score. An alternative could be treating answering "I don't know" to those answerable questions as correct and computing the overall accuracy on the full dataset. Also, why the IK-IK rate for Mistral-7B-Instruct-v0.1 drops drastically compared to IDK-prompting? This is a bit concerning as it indicates after Adaptive Contrastive Learning, the model may not be useful anymore.
>
> **Response**
>
> - **For the definition of evaluation metrics:** we apologize for confusing you due to unclear definition of evaluation metrics. In fact, the IK-IK Rate (I know what I know rate) represents the proportion of questions answered correctly by the model out of all questions. The IK-IDK Rate (I know what I don’t know rate) represents the proportion of questions that the model correctly refuses to answer, out of all questions. **Therefore, the denominators of the IK-IK Rate and IK-IDK Rate are the same. We have clarified this in Section 4.3 of our revised paper**. Furthermore, the Truthful Rate is the sum of the IK-IK Rate and IK-IDK Rate. It represents the proportion of questions for which the model provides truthful responses. *The higher the value of Truthful rate, the clearer the model’s perception of what it knows and does not know, which also indicates a higher level of truthfulness.*
> - **For the IK-IK Rate for Mistral-7B-Instruct-v0.1:** Considering our used three evaluation metrics, the higher these three metrics are, the better. Among these metrics, we argue that the Truthful Rate is the most important one because it indicates the probability that users will receive a truthful response. Therefore:
>    - From the perspective of overall performance, **Adaptive Contrastive Learning has obvious advantages over IDK-prompting and IDK-SFT, which indicates that Adaptive Contrastive Learning will allow the model to produce more truthful responses**, which is relatively more user-friendly, because users certainly don’t like the model not knowing what they know (i.e., catastrophic knowledge forgetting) or not knowing what they don’t know (i.e., knowledge hallucination).
>    - Looking at the IK-IK Rate and the IK-IDK Rate separately, **we think that we should not only see the decrease in the IK-IK Rate, but also the increase in the IK-IDK Rate, and the increase in the IK-IDK Rate is greater than the decrease in the IK-IK Rate**. This shows that after the Adaptive Contrastive Learning's optimization, the model has become more cautious because it knows more about what it does not know, and even answers the knowledge it knows more cautiously. Of course, as mentioned in the above point, we think this is friendly to users. After all, at least the model does not answer the hallucination knowledge that it does not know.
>
> > Weakness 2: I am unsure what's the major takeaway of Section 5.3. Does the accuracy under repeated samplings represent the concept of "confidence"? Actually, the paper mentions the term of "confidence" when introducing the motivation of the proposed method. It would be interesting to see whether the proposed method can draw a more rigorous connection of model's confidence in its generation.
>
> **Response**
>
> We sincerely apologize for the confusion caused to you by Section 5.3 and Figure 4. **First of all, we need to clarify that the vertical axis in Figure 4 represents the proportion of samples that fall into a specific accuracy range in the entire data set.** Specifically, in Section 5.3, we used repeated sampling to make both models answer each question in the TriviaQA test set 10 times. We used the number of correct answers out of the 10 responses (i.e., accuracy) to reflect the performance (or confidence) of the model. Therefore, to explain further, for example, in the “Unknown Questions” sub-figure of Figure 4, the bars with the accuracy of 1, their specific values indicate the proportion of questions that the models correctly rejected to answer 10 times, out of the entire dataset. For example again, in the “Known Questions” sub-figure of Figure 4, the bars with the accuracy of 0.8, their specific value indicate the proportion of questions that the model answered correctly 8 out of 10 times, out of all questions. **We have rewritten Section 5.3 to eliminate potential confusion for readers.**

---

> ### Author Response · Authors · 2024-11-20
> **Response to reviewer BPiT (part 2)**
>
> > Weakness 3: The presentation for the paper can be further improved. See my detailed comments below.
>
> **Response**
>
> Thank you very much for your valuable suggestions, which can make the readability of our paper better, and we have revised them one by one in the updated version according to your suggestions. **Please note that you can see the updated version we uploaded on the OpenReview website.** *To make it more convenient for you to review our revised version, we have highlighted our modified content in blue.*
>
> > Question 1: In Table 1, it's better to use "IDK" instead of "idk" to be accord with the main paper.
>
> **Response**: Yes, we have changed to use 'IDK' in Table 1.
>
> > Question 2: I suggest rewriting Section 5.2 as the current writing is a bit confusing, even though I can get the meaning by reading Table 3. For example, for the first bullet point, "With" should be "Without" based on my understanding.
>
> **Response**: We sincerely apologize for the confusion caused to you by Section 5.2, especially Table 3. We have optimized the writing of Section 5.2 and Table 3 in our revised paper.
>
> > Question 3: I am wondering if the notations of the paper can be improved. For example, using $IDKIK$ as subscript in equations looks weird.
>
> **Response**: Thank you again for your suggestions. We have simplified the notations of our paper. For example, we use ${Q_1}$, ${Q_2}$, ${Q_3}$ to represent Quadrant-1, Quadrant-2, Quadrant-3 in the revised paper.

---

> > ### Comment · Reviewer_BPiT · 2024-11-22
> >
> > I thank the authors for the detailed response. I updated my score to 6 as the revised version resolved many of my previous questions.

---

### Official Review · Reviewer_tNNV · 2024-11-04

**Soundness:** 3
**Presentation:** 3
**Contribution:** 3
**Rating:** 6
**Confidence:** 4

**Summary:**

In this paper, the authors tackle the issue of hallucination, specifically factual errors, in Large Language Models (LLMs) by introducing a novel knowledge boundary delineation method and an Adaptive Contrastive Learning strategy. This approach aims to enhance the model's ability to accurately represent and refine its knowledge, maintain known facts, consolidate uncertain knowledge, and forget incorrect information. Through experiments on both in-distribution and out-of-distribution datasets, the authors demonstrate a significant improvement in the models' Truthful rate, bolstering the validity of their proposed methods and offering valuable insights for future research in enhancing the reliability and honesty of LLMs.

**Strengths:**

The authors propose a new knowledge representation method for LLMs, which assists the model in better refining its own knowledge scope, enhances the model’s honesty and alleviates the model’s hallucination problem through a new knowledge boundary division.

The  authors design an Adaptive Contrastive Learning strategy, through which the model can maintain its known knowledge, consolidate the known but uncertain knowledge, and forget the unknown knowledge, which improves the validity and honesty of the model’s responses.

**Weaknesses:**

The experiments in the paper are not solid enough, only on LLAMA2-7B, and more different models such as Qwen2.5 series should be considered to enhance the said reliability of the experimental results. And the effect on different size models should also be considered and analyzed, such as 13B and above.

The baseline in the paper is too simple and lacks comparison with other up-to-date approaches to knowledge boundary identification.

**Questions:**

Since the use of the proposed method enables the model to distinguish knowledge boundaries effectively, it would be interesting to study the performance of the model after using the RAG technique. For example, comparison experiments with paper [Enhancing Noise Robustness of Retrieval-Augmented Language Models with Adaptive Adversarial Training](https://aclanthology.org/2024.acl-long.540) (Fang et al., ACL 2024) could be added.

---

> ### Author Response · Authors · 2024-11-20
> **Response to reviewer tNNV**
>
> Thank you for your valuable feedback and for pointing out areas for improvement.
>
> > Weakness: The experiments in the paper are not solid enough, only on LLAMA2-7B, and more different models such as Qwen2.5 series should be considered to enhance the said reliability of the experimental results. And the effect on different size models should also be considered and analyzed, such as 13B and above.
> The baseline in the paper is too simple and lacks comparison with other up-to-date approaches to knowledge boundary identification.
>
> **Response**
>
> - While our primary model was indeed LLaMA-2-7B, it was not the only model we used. We also conducted experiments with the Mistral-7B-Instruct-v0.1 and obtained promising results.
>
> - We agree that evaluating different model sizes is crucial. **Given our current resources and timeline, we performed additional experiments on LLaMA-2-13B-Chat**. The results are summarized in the following table:
>
>     |                   | LLaMA-2-13B-Chat      |       |          |
>     |-------------------|-----------------------|-------|----------|
>     |                   | TriviaQA              |       |          |
>     |                   | IK-IK                 | IK-IDK| TRUTHFUL |
>     | IDK-Prompting     | 37.7                  | 31.6  | 69.3     |
>     | IDK-SFT           | 32.8                  | 41.3  | 74.1     |
>     | IDK-SFT-Adpt-Ctr       | 37.8               | 41.1  | 78.9     |
>
>   From the table, we can observe that our method remains competitive even with larger model sizes. **These findings have be updated in Appendix C.2 of our revised paper**.
>
> - We acknowledge the importance of testing other models like Qwen2.5. Unfortunately, due to its recent release, we could not include it in the current version of our paper on time. However, we fully agree with your suggestion and plan to incorporate Qwen2.5 in future experiments to validate our approach across different models.
>
> > Question: Since the use of the proposed method enables the model to distinguish knowledge boundaries effectively, it would be interesting to study the performance of the model after using the RAG technique. For example, comparison experiments with paper Enhancing Noise Robustness of Retrieval-Augmented Language Models with Adaptive Adversarial Training (Fang et al., ACL 2024) could be added.
>
> **Response**
>
> Regarding the suggestion of incorporating the RAG technique, we have used a benchmark RAG-Bench publicly available from the paper mentioned above[1]. RAG-Bench provides specific contexts for each query for the model to reference and respond to. In this experimental setup, we input the context alongside the query into the model (denoted as ``with RAG" in the table) and compared the results with those obtained without using context.
>
> |                   | RAG-Bench       |       |          |
> |-------------------|-----------------------|-------|----------|
> |                   | IK-IK                 | IK-IDK| TRUTHFUL |
> | IDK-Prompting     | 59.7                  | 0.1   | 59.8     |
> | IDK-Prompting with RAG | 63.0                | 5.40  | 68.4     |
> | IDK- SFT          | 60.3                  | 1.90  | 62.2     |
> | IDK-SFT with RAG     | 66.1                  | 3.30  | 69.4     |
> | IDK-SFT-Adpt-Ctr       | 44.5                | 22.4  | 66.9     |
> | IDK-SFT-Adpt-Ctr with RAG | 58.4              | 12.8  | 71.2     |
>
> As evident from the results, all methods benefit from the addition of RAG contexts, and our fine-tuned models exhibit a notable performance boost of 1.8 compared to SFT when utilizing RAG. This demonstrates that RAG enhances model outcomes and that our fine-tuning method integrates effectively with RAG. **We have updated these RAG-related experimental results in Appendix C.3 of our revised paper.**
>
> Reference:
> [1] [Enhancing Noise Robustness of Retrieval-Augmented Language Models with Adaptive Adversarial Training](https://aclanthology.org/2024.acl-long.540) (Fang et al., ACL 2024)

---

> > ### Comment · Reviewer_tNNV · 2024-11-21
> > **Response to authors**
> >
> > Thanks to the authors for their response, the additional experimental results are useful.
> > For the new experiments in the RAG scenario I found that the IK-IDK of both IDK-Prompting and IDK-SFT is very low (even 0.1), and then RAG has a certain increase in the IK-IDK of IDK-Prompting and IDK-SFT, but there is a significant decrease in the IK-IDK of IDK-SFT-Adpt-Ctr.
> > Can these phenomena be analyzed? After that I will consider raising my score appropriately.

---

> > > ### Author Response · Authors · 2024-11-21
> > > **Further response to reviewer tNNV**
> > >
> > > Thank you very much for your insightful and valuable suggestions. We hope the response below addresses your concern.
> > >
> > > Because we want to respond to you as soon as possible during the rebuttal phase, we did not carefully design the prompts of IDK-Prompting on the RAG-Bench.
> > > After closely analyzing your comments and our results, we agree that the IK-IDK rate for IDK-Prompting should not be as low as initially observed.
> > > **Therefore, we conducted further experiments with different refined prompts and found that the performance of IDK-Prompting is quite sensitive to how the specific prompt is phrased, which is also a potential weakness of the IDK-Prompting baseline.**
> > > Below are the results when we used a more straightforward and refined prompt:
> > >
> > > |                   | RAG-Bench       |       |          |
> > > |-------------------|-----------------------|-------|----------|
> > > |                   | IK-IK                 | IK-IDK| TRUTHFUL |
> > > | IDK-Prompting     | 59.7                  | 0.1   | 59.8     |
> > > | IDK-Prompting with RAG | 63.0                | 5.4  | 68.4     |
> > > | IDK-Prompting (refined prompt)     | 47.4    | 11.3   | 58.7    |
> > > | IDK-Prompting with RAG (refined prompt)     | 62.5    | 5.8   | 68.3    |
> > > | IDK-SFT          | 60.3                  | 1.9  | 62.2     |
> > > | IDK-SFT with RAG     | 66.1                  | 3.3  | 69.4     |
> > > | IDK-SFT-Adpt-Ctr       | 44.5                | 22.4  | 66.9     |
> > > | IDK-SFT-Adpt-Ctr with RAG | 58.4              | 12.8  | 71.2     |
> > >
> > > From the latest results, it can be seen that IDK-Prompting and IDK-SFT-Adpt-Ctr show a decreasing trend in IK-IDK after combining with RAG, while IDK-SFT shows an increasing trend in IK-IDK after combining with RAG. The reasons for this phenomenon are as follows:
> > >
> > > - It is intuitive and reasonable that IDK-Prompting and IDK-SFT-Adpt-Ctr exhibit a decreasing trend in IK-IDK when combined with RAG, **as the model naturally tends to answer questions more directly and reduce refusal-to-answer after being input with more additional contextual information from RAG.** Therefore, from the perspective of metrics, after adding RAG to these two methods, IK-IDK decreased.
> > > - The phenomenon of the increase in IK-IDK after adding RAG to IDK-SFT can only indicate an increase in correct refusal-to-answer, but this does not conflict with the above analysis, **as the overall refusal-to-answer include both correct and wrong ones**. After careful observation, we found that the overall rejection rate of IDK-SFT is about 10%, while the overall rejection rate of IDK-SFT-RAG is about 4%. **This indicates that the overall rejection rate of IDK-SFT still decreases after adding RAG, and the main reason for the increase in IK-IDK is that a considerable portion of refusal-to-answer in IDK-SFT is wrong**.
> > >
> > > Thank you again for your insightful comments, which have deepened our understanding of the methods discussed in our paper. **We have updated the above analyses to Appendix C.3 of our revised paper.**

---

> > > > ### Comment · Reviewer_tNNV · 2024-11-25
> > > > **Response to authors**
> > > >
> > > > Thanks to the author's further response, I have increased my score to 6.

---

> ### Author Response · Authors · 2024-11-24
> **Friendly reminder to reviewer tNNV**
>
> Thank you very much for your insightful suggestions and comments. **Based on your advice, we have analyzed the experimental phenomena under the RAG scenario and have updated our revised paper accordingly**. We hope these analyses can address your concern.
>
> Considering that the rebuttal discussion phase of ICLR is about to end, we sincerely hope to receive your further feedback so that we have the opportunity to continue our discussion with you. Once again, thank you for your hard work and selfless help.

---

### Author Response · Authors · 2024-11-20
**General response to reviewers**

Dear Reviewers,

Thank you for your hard work and valuable comments, which make our paper better. According to ICLR's rebuttal policy, **we have revised our paper according to your comments and uploaded it to OpenReview**. In addition, for your convenience, **we have highlighted the revisions in blue in the revised paper**. Thank you again for your work and look forward to further communication with you.

Authors of Paper 8970

---

### Author Response · Authors · 2024-12-02
**Appreciation for the Efforts of Reviewers and Area Chair**

Dear Reviewers and Area Chair,

As we approach the end of the rebuttal phase, we would like to express our sincere gratitude for your time, effort, and insightful comments on our paper. Your constructive comments and suggestions have been invaluable in helping us improve our work.

During the rebuttal process, we have strived to ensure that we addressed all the concerns raised by the reviewers, and have made corresponding revisions to our paper as suggested. **We are more than happy to know that changes have significantly improved the quality and clarity of our work, which is reflected in the positive scores that all reviewers have given to our paper. Particularly, we are grateful for the consistent approval shown in the raised scores from Reviewer tNNV and Reviewer BPiT.**

Your unanimous recognition of the value and contribution of our research is greatly appreciated. You motivate us to strive for further excellence. Once again, thank you for your dedication and the pivotal role you play in maintaining the high standards of the ICLR conference.

Best regards,

Authors of Paper 8970

---

### Meta-Review · Area_Chair_FVYT · 2024-12-25

**Metareview:**

This paper addresses the hallucination problem in Large Language Models (LLMs) by introducing an Adaptive Contrastive Learning strategy. By imitating human learning, the method constructs positive and negative samples dynamically based on the model's knowledge mastery, enabling LLMs to consolidate correct knowledge, discard incorrect knowledge, and recognize gaps in their understanding. Experiments demonstrate significant improvements in truthfulness and reliability compared to existing methods.

Strength: The paper presents an innovative Adaptive Contrastive Learning strategy for addressing the problem. This approach effectively helps LLMs refine their knowledge boundaries, enhance response validity and honesty, and mitigate hallucination issues. The method’s strong experimental results, well-motivated design, and novel integration of contrastive learning demonstrate its effectiveness across diverse datasets, making it a valuable contribution to improving LLM reliability.

Weakness: Reviewers mentioned a few weakness points such as low experimental coverage, with evaluations performed on a narrow range of models and datasets, etc.

**Additional Comments On Reviewer Discussion:**

Most of the concerns raised by reviewers have been addressed during the rebuttal phase (such as lack of depth in the baseline comparisons, insufficient exploration of the balance and interactions between the proposed loss functions, and insufficient discussion around "I know that I don't know"). All reviewers agree that this paper has made positive contribution to the area and acknowledge that the authors have addressed the majority of their questions. There are some remaining points such as missing experiments on additional foundation models. I believe these results would be useful but not essential. Overall, this paper made a novel contribution by developing an contrastive learning method to solving the LLM hallucination problem, which could be inspiring for the community. So I would recommend acceptance of this paper.

---

### Decision · Program_Chairs · 2025-01-22

Accept (Poster)